# GIST: Generating Image-Specific Text for Fine-grained Object Representations

## Abstract

Recent models pre-trained on image-text pairs can learn rich vision-language representations that improve downstream tasks, such as image classification. However, because of the absence of paired text/image descriptions in many fine-grained domains, such as dermatology, it is difficult to fine-tune these models for specific downstream tasks. In this work, we propose GIST – a method for generating *image-specific fine-grained* text descriptions from image-only datasets. Our findings include 1) prompting a pretrained large language model with *domain-specific* prompts generates diverse fine-grained text descriptions that capture inter-class and intra-class differences, 2) using a pretrained vision-language model to match each training image to the most relevant text descriptions generates image-specific image-text pairs, and 3) summarizing the matched text using a large language model prior to fine-tuning the image encoder improves the utility of the learned representations. We demonstrate the utility of GIST by fine-tuning vision-language models on the GIST-generated image-text pairs to learn an aligned vision-language representation space. We evaluate this learned representation space in full-shot and few-shot scenarios across four diverse fine-grained classification datasets, each from a different domain. Our method achieves an average of $1.1\%$ improvement in accuracy over the existing state-of-the-art image-text classification method and $4.1\%$ improvement in accuracy over CLIP linear probes on full-shot datasets. Our method achieves similar improvements across few-shot regimes. Code will be made publicly available upon publication.

## 1 Introduction

The recent development of foundation models trained on image-text pairs has led to impressive performance across vision and language tasks such as zero-shot classification, image generation, and image captioning (Radford et al., 2021; Ramesh et al., 2021; Wang et al., 2022). Fine-tuning these models for domain-specific tasks requires image-text pairs, which are often costly to obtain (Zhou et al., 2022a;b). In particular, fine-tuning these models for fine-grained image classification requires the construction of image-specific text prompts that differentiate visual features between subcategories of objects, e.g., between species of birds. Trained image captioning methods, such as GIT (Wang et al., 2022), can generate descriptive captions for coarse-object-level interactions, but often are not able to generate the details needed for fine-grained object representations (Fig. 1).

Large language models (LLMs) provide an opportunity for generating fine-grained text. Models, such as GPT (Brown et al., 2020; OpenAI, 2023), contain rich prior knowledge about both coarse and fine-grained classes Lee et al. (2020); Touvron et al. (2023); Gu et al. (2021). Large language models are able to generate fine-grained text descriptions when prompted about a specific class. Unfortunately, these descriptions are not specific to a particular image. Recent works have explored how to bridge this gap to use text generated by GPT to improve few-shot image classification (Yang et al., 2023; Zhang et al., 2023). These works use generic prompt templates, such as "What does a *cardinal* look like?", to generate class-specific text descriptions. We show that instead of using generic templates, constructing *domain-specific* prompts that highlight potential sub-class differences can lead to more specific text descriptions that differentiate visual features of fine-grained classes. By automatically matching each training image to specific captions, we generate diverse image text pairs to fine tune multi-modal networks for fine-grained classification tasks.

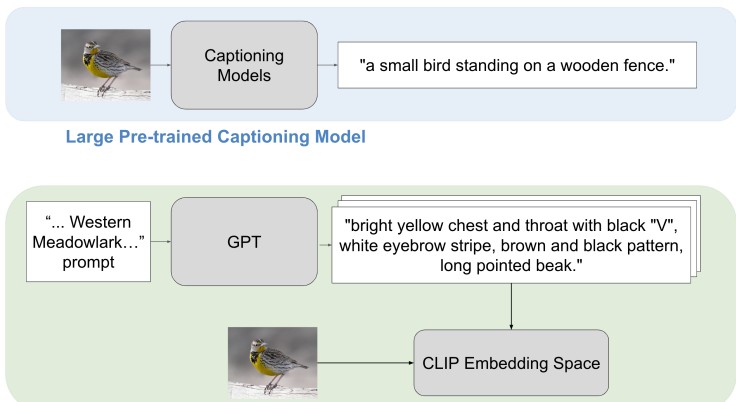

Figure 1: Large pretrained vision-language models (top), such as GIT, perform well on tasks involving coarse-level descriptions, but often don't generalize well to fine-grained tasks. Our method (bottom), GIST, generates fine-grained class-specific text descriptions. It then uses CLIP to match each image to the closest label-preserving text. This results in fine-grained image-text pairs where the text describes specific visual features in the image.

We introduce GIST (Generating Image-Specific Text), a novel approach for generating fine-grained, image-specific text descriptions for *any-shot* fine-grained classification. We first use an LLM to generate descriptive and diverse class-specific captions to cover the possible range of visual appearances of each class. We then use a pretrained vision-language model to match each training image to the closest label-preserving text descriptions so that the matched text specifically describes the object of interest in the image. Our insight is that many domains can have intra-class differences (e.g., male and female birds of the same species can appear differently) so it is important to prompt the LLM to capture this diversity and then use a vision-language model to align the images and text within each class. Once we have aligned image-text pairs, we use the LLM to summarize the long text descriptions into a concise format well suited for fine tuning CLIP. We fine-tune CLIP on our GIST image-text pairs to learn a better aligned vision-language representation space for downstream applications, such as classification.

We evaluate our method against recent vision-language classification methods. We show that our model improves upon state-of-the-art results on four diverse datasets across many different training scenarios. Unlike other methods that are optimized for either full or few-shot classification, our method is consistently better than existing methods on both scenarios.

The key contributions in this paper are:

- We introduce GIST, a method for generating fine-grained, image-specific text. To our knowledge, we are the first to use language priors and image-text foundation models to generate image-specific captions for fine-grained datasets.
- We demonstrate that fine-tuning CLIP on our GIST image-text pairs outperforms recent vision-language classification methods on four fine-grained datasets in both full-shot and few-shot classification regimes.
- We provide in-depth analysis on GIST by comparing to existing captioning methods, comparing to a visual-grounding approach, and studying the impacts of CLIP model, caption length, and number of captions on our classification performance.
- We provide a new fine-grained image classification dataset, Fitzpatrick40.

## 2 RELATED WORK

### 2.1 VISION-LANGUAGE MODELS

Models pretrained on large-scale multimodal datasets provide data representations that are transferable to many tasks and domains (Joulin et al., 2016; Li et al., 2017; Desai & Johnson, 2021; Sariyildiz et al., 2020). CLIP (Radford et al., 2021) is pretrained using self-supervised contrastive

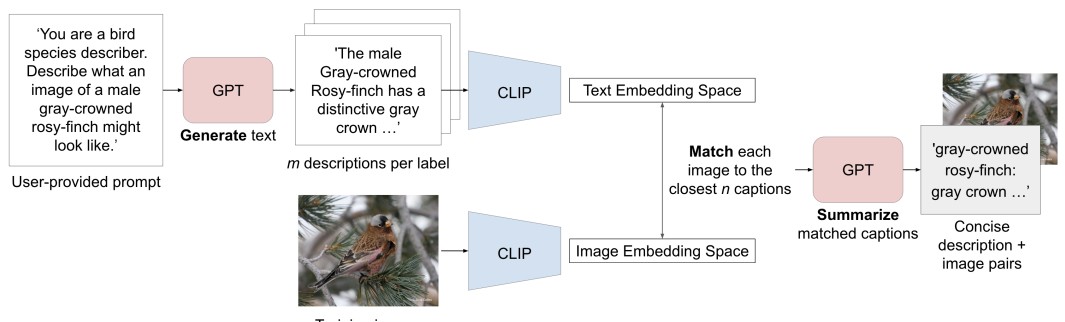

Figure 2: GIST overview. We leverage prior domain knowledge captured by large language models (e.g., GPT) to generate multiple text descriptions for each label. We use CLIP to match each training image to the closest $n$ text descriptions with the same label. We then use GPT to summarize each of the matched captions. We use the matched text-image pairs to train downstream models.

learning on a dataset of approximately 400 million image-text pairs and has shown strong performance for both zero-shot learning and other downstream tasks. Other related approaches achieve similar performance with larger and noisier datasets (Jia et al., 2021), or fine-tune CLIP to adapt it to out-of-domain datasets (Kumar et al., 2022; Miller et al., 2021; Wortsman et al., 2022; Zhai et al., 2022). Still other methods improve the efficacy of CLIP training by incorporating self-supervised training techniques (Mu et al., 2022; Li et al., 2021; 2023), re-writing captions for pretraining images (Fan et al., 2023), and generating captions for images using off-the-shelf captioning models (Nguyen et al., 2023).

Large-scale captioning models (Wang et al., 2022; Yu et al., 2022) leverage image-text pairs to learn a shared representation space for generating text from images. However, we show in our experiments that these captioning models do not perform well on fine-grained classes. Large-scale image generation models such as DALL-E (Ramesh et al., 2021), Imagen (Saharia et al., 2022), and stable diffusion (Rombach et al., 2022) are trained on large-scale image-text data in order to learn to generate photo-realistic images from text. In the few-shot learning literature, DALL-E has been shown to help by augmenting small image datasets with synthetic images (Zhang et al., 2023). This works well for Imagenet (Russakovsky et al., 2015) classes, but we show in our experiments the generated images are not specific or accurate enough for fine-grained domains, such as identifying skin diseases.

## 2.2 LARGE LANGUAGE MODELS

Large language models trained with huge quantities of textual data have achieved promising performance on many NLP tasks (Devlin et al., 2018; Lee et al., 2020; Brown et al., 2020; Touvron et al., 2023; Chowdhery et al., 2022). GPT (Brown et al., 2020; OpenAI, 2023) is a self-supervised pretrained large language model with 175 billion parameters. GPT can be used for a variety of language tasks, including text generation. GPT can generate accurate, detailed descriptions for many different domains given the appropriate prompt. We use the strong knowledge embedded in GPT to generate text descriptions, and show that these text descriptions carry more useful information than generated image captions from models such as GIT Wang et al. (2022). Our method pairs these text descriptions with images to improve classification over image-only methods.

## 2.3 FEW-SHOT CLASSIFICATION USING FOUNDATION MODELS

Recent methods leverage foundation models for improving few-shot classification. A few methods improve CLIP zero-shot or few-shot performance by constructing text prompts from the training image labels (e.g. "a photo of *class name*") (Udandarao et al., 2022; Lin et al., 2023; Goyal et al., 2023). LaBo (Yang et al., 2023) uses GPT-3 to generate candidate concepts for each class and aligns these text concepts to the images using CLIP. We show in our experiments that generating full text descriptions using GPT improves classification accuracy more than using label names or concepts. The Prompt, Generate, then Cache method (Zhang et al., 2023) uses generated text from

GPT and synthetic images from DALL-E (Ramesh et al., 2021) to supplement few-shot learning. We show in our experiments that DALL-E is not able to generalize to many domains, and that using the unrealistic synthetic images can hurt classification performance. LaCLIP Fan et al. (2023) uses LLMs for text augmentation to improve CLIP pre-training on image-text datasets. Their experiments demonstrate that their language augmented CLIP models achieve better performance than CLIP on downstream tasks such as classification. Unlike LaCLIP, our method generates text captions rather than augmenting them; therefore our method can be used on image-only datasets rather than requiring image-text datasets.

A concurrent paper (Maniparambil et al., 2023) to our work uses GPT-4 and a pre-trained CLIP model to improve zero-shot and few-shot classification. Their method similarly generates $N$ text descriptions per class. Instead of matching images to text descriptions and fine-tuning CLIP, their method averages over the $N$ text embeddings per class to get a representative embedding per class. They perform classification by computing cosine distance between the image embeddings and each averaged class text embedding to improve zero-shot CLIP performance. In contrast, our generated image-text pairs allow us to fine tune CLIP to achieve higher overall classification accuracy.

## 3 METHOD

Our key contribution is a method for building an image encoder for fine-grained image classification tasks. In this section, we first introduce the fine-grained classification problem and then describe how we use prior knowledge contained in large language and vision-language models for generating image-specific text descriptions (Figure 2).

### 3.1 PROBLEM SET-UP

Consider a set of images $\{x_1, ..., x_n\}$ that have corresponding fine-grained $k$-class classification labels $\{y_1, ..., y_n\}$, where each $y_i \in \{c_1, ..., c_k\}$. We aim to learn an embedding that facilitates building an image classifier $f_\theta$, to solve $y = f_\theta(x)$, given limited examples $n$.

We harness prior knowledge encoded in pretrained large language model $llm(d)$ and pretrained vision-language model $vl(d, x)$, for text description $d$ and image $x$. We use $vl_L(d)$ to represent the language encoder and $vl_V(x)$ to represent the vision encoder of the vision-language model.

A pivotal insight is to define the learned model as
$$y = f_\theta(x) = g_{\theta_g}(h_{\theta_V}(x)), \tag{1}$$
where $\theta = \{\theta_g, \theta_V\}$, and $g_{\theta_g}(\cdot)$ is a small classifier with few parameters $\theta_g$, applied to the embedding space learned by the *fixed* image encoder model $h_{\theta_V}(\cdot)$. We aim to first learn a relevant image encoder $h_{\theta_V}(\cdot)$ separately from the classification model.

We learn this image encoder $h_{\theta_V}(\cdot)$ by aligning images and associated text descriptions $\{x_i, \mathcal{D}_i\}$ where $\mathcal{D}_i$ represents a list of associated text descriptions for image $x_i$. We align the images and generated text descriptions in a contrastive learning manner, similar to CLIP training. We optimize the objective for the image and language encoders $h_{\theta_V}$ and $h_{\theta_L}$, respectively,

$$L(\theta_V | \{x_i, \mathcal{D}_i\}) := \sum_{i=1}^{n} \sum_{d_i \in D_i} -\log \frac{\exp(h_{\theta_V}(x_i) * h_{\theta_L}(d_i))}{\sum_{j=1}^{n} \sum_{d_j \in D} \exp(h_{\theta_V}(x_i) * h_{\theta_L}(d_j))}$$
$$+ \sum_{i=1}^{n} \sum_{d_i \in D_i} -\log \frac{\exp(h_{\theta_V}(x_i) * h_{\theta_L}(d_i))}{\sum_{j=1}^{n} \sum_{d_j \in D} \exp(h_{\theta_V}(x_j) * h_{\theta_L}(d_i))},$$

where $\mathcal{D} := \cup \mathcal{D}_i$ and $\theta_V$ and $\theta_L$ represents the image and text encoder parameters, respectively. The resulting $h_{\theta_V}$ can be used as the frozen image encoder for the fine-grained classification task. We next describe how to use GIST to generate image-associated text descriptions $\{x_i, \mathcal{D}_i\}$.

### 3.2 VISION AND CLASS PRIORS VIA LANGUAGE MODELS

We generate image-associated text descriptions $\{x_i, \mathcal{D}_i\}$ using priors captured in existing large language models.

We first synthesize a set of $m$ description candidates $\mathcal{D}_{c_j} = \{d_{c_j,m}\}_{m=1}^{M}$ for each class in $\{c_1, ...c_k\}$. We employ a stochastic large language model

$$d_{c_i,m} = llm(\text{prompt}(c_i), s),$$

for random seed $s$. We describe prompt$(\cdot)$ below and in more detail in the Appendix.

We then form a set of $t$ descriptions for each image $x_i$, $\mathcal{D}_{x_i} = \{d_{x_i,j}\}_{j=1}^{t}$ by using priors captured in existing vision-language models. We form $\mathcal{D}_{x_i}$ by choosing the top-$t$ descriptions in $d \in \mathcal{D}_{y_i}$ that minimize the cosine similarity:

$$dist(vl_V(x), vl_L(d)).$$

Next, we use $llm(\cdot)$ to summarize the matched text descriptions into a more concise form. Following summarization, we append the corresponding class name to each concise caption to generate $\{x_i, \mathcal{D}_i\}$.

### 3.3 MODEL DETAILS

We use GPT-3 as the pretrained prior language model $llm(\cdot)$ and the ViT-L/14@336px CLIP model as the prior vision-language model $vl(\cdot)$. We initialize the image and language encoders $h_{\theta_V}(\cdot)$ and $h_{\theta_L}(\cdot)$ using CLIP image and language encoders $vl_V(\cdot)$ and $vl_L(\cdot)$. Figure 3 illustrates the fine-tuning process.

### 3.4 PROMPTING CLASS DESCRIPTIONS

We generate sample descriptions by creating LLM prompts that include *domain-specific* characteristics. These help substantially for downstream matching of captions to images. For each domain, we create *one* prompt template that identifies potential intra-class differences in that domain. This prompt template can then be used across classes and datasets within that domain, amortizing the small amount of upfront human labor. For example, the template might highlight the fact that male and female birds of the same species can appear differently ("Describe what an image of a *gender species* might look like") or that dermatology diseases can appear on different parts of the body ("Describe what an image of *disease* might look like on a person's *body part*"). We can then prompt GPT-3 while iterating through a set list of options for the intra-class difference category. This does not require in-depth domain-specific knowledge as we iterate through the same list of options for every label and do not need to know which classes have intra-class differences. We provide a list of the prompts used for each of the domains in our experiments in Appendix Section A.3.

## 4 EXPERIMENTS

In this section, we provide training details and evaluate GIST on fine-grained classification. We first compare GIST to several recent state-of-the-art benchmarks on any-shot classification performance for four fine-grained classification datasets. We then evaluate all methods using different CLIP models. We also analyze how different GIST hyperparameter choices, including the number of matched captions and the length of matched captions, affects downstream classification performance.

### 4.1 DATASETS

We evaluate on four fine-grained classification datasets:

- **CUB200-2011** (Wah et al., 2011): The CUB200-2011 dataset has 11,788 labeled images of 200 different bird species. We use the published train/test split and separate 10% of the training set as a validation set.
- **Flowers102** (Nilsback & Zisserman, 2008): The Flowers102 dataset has 8,189 labeled images of 102 different flower species. We use the published train/val/test split.
- **FGVC-Aircraft** (Maji et al., 2013): The FGVC-Aircraft dataset has 10,000 labeled images of 100 different types of aircrafts. We use the published train/val/test split.
- **Fitzpatrick40** – a cleaned subset of Fitzpatrick17k (Groh et al., 2021): The Fitzpatrick17k dataset has 16,577 labeled images for 114 different skin diseases. The Fitzpatrick17k dataset includes

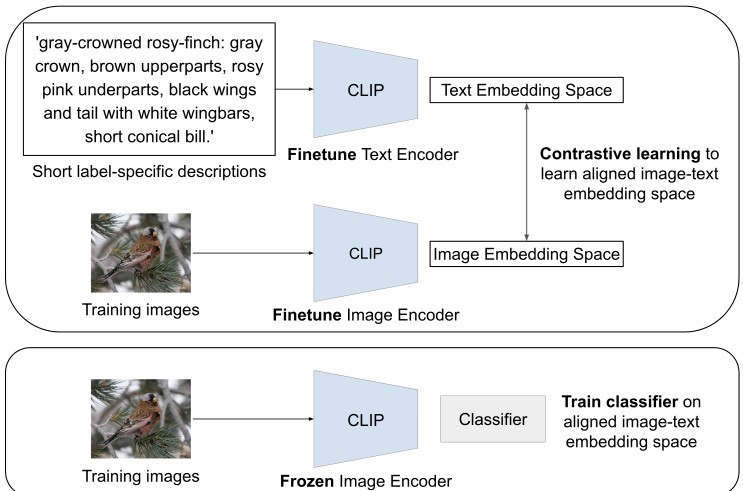

Figure 3: Overview of proposed training of image encoder and classifier. We use a contrastive strategy, fine-tuning the vision-language model from a CLIP initialization, using paired training images and image-specific captions generated with the proposed GIST model. Given a learned aligned image-text embedding space, we freeze the CLIP encoders and train a small classifier on our original data.

erroneous images (Pakzad et al., 2023), and we also find that it has many mislabeled images, where the image label in the dataset does not match the image label from the image's original medical website source. We clean a 40-label subset of the 114-label original dataset. We denote this cleaned dataset as Fitzpatrick40. We include more information about the dataset cleaning process in the Appendix. Fitzpatrick40 includes 2,609 images, which we split into 2,115 training, 222 validation, and 272 test. We provide the cleaned Fitzpatrick40 dataset as an additional contribution of this work.

For all four datasets, we generate *k*-shot datasets by randomly sampling *k* training images from each label. We report the 5-shot results in the main paper and report the 3-shot and 1-shot results in the Appendix.

## 4.2 CLASS DESCRIPTIONS

For the Fitzpatrick40, Flowers102, and FGVC-Aircraft datasets, we generate text descriptions using GPT-3 (Brown et al., 2020). For the CUB200-2011 dataset, we found that GPT-3 did not generate captions with accurate information. For this dataset, we generate text descriptions using GPT-4 (OpenAI, 2023). We generate between 20 and 60 different class-specific captions, depending on the dataset.

## 4.3 BASELINES

**Image-only Baselines.** We train a **ResNet50** network (He et al., 2016) for 600 epochs using SGD with a learning rate of 0.001, batch size of 64, momentum of 0.9, and 1e-4 weight decay. We also train a second ResNet50 with class re-weighting (**ResNet-RW**) to compensate for class imbalance. We use a weighted random sampler to sample from each class with equal probability during training.

**Multi-modal Baselines.**

- **Zero-shot CLIP (CLIP ZS)** and **Linear-probe CLIP (CLIP LP)** (Radford et al., 2021) Since CLIP is trained to align image-text datasets, we perform zero-shot classification using the pre-trained CLIP model. Given $y \in Y$ class names, we use standard templates (e.g. "a photo of a ...") to construct text descriptions for each class $y$. We use the average embedding, $\overline{vl_L(y)}$, of the descriptions as the class embedding. We then perform zero-shot classification CLIP-ZS for image $x$ by finding the most similar class embedding to image embedding $vl_V(x)$ using cosine similarity

| Methods | Fitzpatrick40 | | CUB200 | | Flowers102 | | FGVC | |
| | Full-shot | 5-shot | Full-shot | 5-shot | Full-shot | 5-shot | Full-shot | 5-shot |
|---|---|---|---|---|---|---|---|---|
| ResNet | 67.30 (2.87) | 31.62 (3.69) | 79.08 (0.56) | 49.83 (0.47) | 98.54 (0.24) | 81.35 (0.35) | 69.79 (0.79) | 22.31 (0.54) |
| ResNet-RW | 70.95 (2.84) | - | 79.46 (0.55) | - | 69.79 (0.79) | - | 69.33 (0.77) | - |
| CLIP ZS | 2.94 (1.06) | 2.94 (1.06) | 63.29 (0.63) | 63.29 (0.63) | 76.64 (0.84) | 76.64 (0.84) | 32.18 (0.83) | 32.18 (0.83) |
| CLIP LP | 71.19 (2.84) | 41.49 (2.99) | 86.22 (0.45) | 76.41 (0.57) | 99.36 (0.17) | 97.48 (0.32) | 65.43 (0.83) | 45.13 (0.86) |
| LaBo | 57.35 (0.19) | 32.17 (0.25) | 83.00 (0.03) | 73.66 (0.07) | 99.26 (0.03) | 96.06 (0.02) | 64.16 (0.15) | 45.20 (0.02) |
| CaFo | - | 40.32 (0.63) | - | 69.48 (0.28) | - | 95.40 (0.68) | - | 43.16 (1.15) |
| FLYP | 73.16 (2.70) | 42.18 (2.98) | 86.41 (0.47) | 78.68 (0.52) | 99.47 (0.13) | 97.80 (0.33) | 71.70 (0.75) | 48.07 (0.87) |
| **GIST (Ours)** | **75.77 (2.67)** | **46.96 (3.13)** | **87.64 (0.44)** | **79.52 (0.55)** | **99.60 (0.15)** | **97.88 (0.29)** | **72.27 (0.76)** | **49.44 (0.86)** |

Table 1: Fine-tuning with GIST captions achieves the best top-1 accuracy for full-shot and 5-shot classification across all datasets. We report the average accuracy and standard deviation over 1000 bootstrapped samples of the test set for full-shot and CLIP-zero shot experiments. For 5-shot experiments, we report the average accuracy and standard deviation over three 5-shot samples.

of the l2-normalized embeddings. For CLIP-LP, we learn a linear classifier on the frozen image embeddings computed using the pretrained image encoder $vl_V(x)$. We train the linear classifier until convergence (500 epochs), using standard SGD with a learning rate of 0.05, batch size of 64, momentum of 0.9, and 1e-4 weight decay. We select the model weights from the epoch with the best validation accuracy to use for experiments.

- **Language in a Bottle (LaBo)** (Yang et al., 2023) We train LaBo, using the provided open source code, on the full and $k$-shot datasets. We use the same CLIP model as in our method and generate text concepts using the same GPT versions as in our method.

- **Cascade of Foundation Models (CaFo)** (Zhang et al., 2023) We train CaFo, using the provided open source code, on each $k$-shot datasets. We do not compare to the full training set results since these are not reported in the original paper, and CaFo was developed for few-shot learning. For the Flowers102 and FGVC datasets, we use CaFo pre-generated DALL-E images and GPT-3 prompts. For the other two datasets, we generate our own DALL-E images and GPT prompts following the CaFo paper process. We use the same GPT versions and CLIP version that we use for GIST. For each k-shot experiment, we report the best score between using 1 DALL-E generated image or using k DALL-E images.

- **Finetune Like You Pretrain (FLYP)** (Goyal et al., 2023) We fine-tune the pretrained CLIP model using class names only with simple templates (e.g. "a photo of a ...") using contrastive training. We then train a linear probe on the learned image encoding and report the classification accuracy.

## 4.4 Fine-Grained Classification Results

**Quantitative Results.** Table 1 summarizes full-shot and 5-shot classification top-1 accuracy results, and we provide 3-shot and 1-shot results in the Appendix. For the full-shot and for the CLIP-ZS experiments, we compute statistics over 1000 bootstrapped samples of the test set. For 5-shot, we report the accuracy and standard deviation on the full test set for each dataset. For each $k$-shot experiment, we average the test results over three samples of $k$ training images.

GIST outperforms all baselines on all datasets, often by substantial margins, for top-1 accuracy in both the full-shot and 5-shot case. Our improvements over the previous state-of-the-art, FLYP, are statistically significant, using a paired t-test. We find a similar trend for top-3 accuracy (Appendix). The Flowers102 and CUB200 datasets have relatively high image-text alignment in the pretrained CLIP embedding space as shown by the CLIP-ZS accuracy. In contrast, the Fitzpatrick40 and FGVC datasets have low CLIP-ZS accuracy, indicating they are less well represented in the pretrained CLIP. GIST is able to achieve better classification performance for all four datasets despite the difference in initial image-text alignment.

**Qualitative Results.** We also analyze the generated data for each method to better understand the quantitative results. Our method, CaFo, and LaBo all use the same GPT models, but each method prompts GPT differently and has different ways of using the generated text. Figure 4 shows randomly selected examples of ground truth images from each dataset, corresponding generated text from each method, and generated DALL-E images (for CaFo). Additional examples can be found

| | Training Images | CaFo DALL-E Images and Generated Text | LaBo Candidate Concepts | FLYP Text | GIST Matched Text (Ours) |
|---|---|---|---|---|---|
| Fitzpatrick40 | | "Becker nevus can typically be identified by its typically large, grey-brown raised patch, often covered by darker hairs. It is often found on the back, shoulder, or chest, and can have a mottled border. It is also usually painless, and cannot be wiped off." | "dark hairs" "hairier than the surrounding skin" "raised irregular shape" | "A photo of becker nevus" | "becker nevus: asymmetrical dark, brown patch" |
| CUB200 | | "The Common Yellowthroat is a small species of warbler bird found in North America. They have a distinctive black 'bandit' mask across their eyes which is more prominent in males, their throat is bright yellow, their belly is white…" | "weighs 7 to 13 grams" "black mask that extends through its yellow eyes" "prefers open habitats such as marshes, fields, and edges of woods" | "A photo of a common yellowthroat" | "common yellowthroat: bright yellow throat and chest, olive-green upperparts, black "bandit" mask, white belly, narrow pointed beak." |
| Flowers102 | | "A photo of a flower sweet william, it has a white color or sometimes pink color flower with five petals. The center of the flower is often yellow color. The leaves are green." | "shaped like a small bell" "beautiful and popular choice for gardens" "very durable flower" | "A photo of a sweet william" | "sweet william: clumps of clusters of small, densely packed, flat-topped flowers with five petals in shades of pink, red, or white." |
| FGVC | | "The ERJ 135 is a regional jet that was manufactured by Embraer. It is a twin-engine aircraft that has a capacity of 37 passengers." | "powered by two turbofan engines mounted on pylons underneath the wings" "designed to connect small cities to larger hub airports" "range of 1,935 miles" | "A photo of a ERJ 135" | "ERJ 135: regional jet aircraft, low-wing configuration, engines mounted on wing pylons, T-tail design, sleek fuselage, seating capacity of about 37 passengers" |

Figure 4: Qualitative examples of dataset training images and corresponding generated data from each method. Compared to other methods, GIST provides concise captions that capture key visual features in the image relevant to classification.

in the Appendix. Our image-text matching and text summarization steps ensure that our matched text is concise and image-specific. In comparison, CaFo generates long text that, while informative, contains extraneous words and might not be specific to the described image. CaFo's DALL-E images convey correct coarse-level details, but often miss or incorrectly generate fine-grained details. For Fitzpatrick40, the DALL-E images often contain class-inconsistent details such as incorrect shape, texture, or color of the disease. We show more examples of the DALL-E images in the Appendix (Figure 9). LaBo generates text concepts for each class, uses a selection function to narrow the set of candidate concepts, and then weights the similarity between classes and each concept. We find that some concepts carry useful information, while others are generic or not descriptive of visual features. For example, the LaBo text description "black mask that extends through its yellow eyes" provides useful visual knowledge of the Common Yellowthroat. However, LaBo text descriptions such as "prefers open habitats such as marshes, fields, and edges of woods" and "beautiful and popular choice for gardens" are less helpful for classifying images of bird or flower species. FLYP creates text by appending "A photo of a" to each class name. FLYP achieves impressive results with their short captions and does not risk having incorrect or irrelevant information like CaFo and LaBo do. However, the FLYP text descriptions are not interpretable and do not capture visual descriptors that describe and differentiate classes. This provides less insight into how the model makes its classification decisions.

## 4.5 METHOD ANALYSIS

We analyze and provide insight into the GIST design choices. We run all ablation studies on Fitzpatrick40 since it is the most difficult few-shot dataset. We expect that our ablation findings would generalize across the other datasets.

| Methods | RN50 | ViT-B/16 | ViT-B/32 | ViT-L/14 | ViT-L/14@336px |
|---|---|---|---|---|---|
| CLIP LP | 31.97 (2.85) | 38.37 (2.93) | 35.88 (2.90) | 42.27 (3.14) | 41.49 (2.99) |
| FLYP | 32.58 (2.84) | 40.35 (3.03) | 37.25 (2.87) | 43.97 (3.05) | 42.18 (2.98) |
| **GIST (Ours)** | **33.40 (2.81)** | **43.91 (3.01)** | **42.08 (2.97)** | **47.45 (3.12)** | **46.96 (3.13)** |

Table 2: GIST-generated captions improves on 5-shot Fitzpatrick40 classification over a selection of different CLIP models. The top-3 GIST matched captions are used for fine-tuning.

**CLIP Model Selection.** We show classification accuracy on the 5-shot Fitzpatrick40 dataset using different CLIP models for the linear probing baseline, the best performing baseline, and GIST. Table 2 shows that performance for all methods can differ substantially depending on the CLIP model. However, GIST consistently outperforms the other methods regardless of the CLIP model used.

| Methods | 1 | 2 | 3 | 4 | 5 |
|---|---|---|---|---|---|
| Fitzpatrick | 44.10 (2.97) | 44.43 (3.04) | **46.96 (3.13)** | 45.15 (3.01) | 46.69 (3.03) |
| CUB200 | **79.52 (0.55)** | 79.14 (0.55) | 79.19 (0.54) | 79.34 (0.54) | 79.25 (0.54) |
| Flowers102 | **97.88 (0.29)** | 97.52 (0.31) | 97.52 (0.31) | 96.96 (0.35) | 97.06 (0.34) |
| FGVC | 49.13 (0.87) | 48.69 (0.86) | 49.33 (0.86) | **49.44 (0.86)** | 48.54 (0.86) |

Table 3: We report average accuracy and standard deviation for 5-shot classification with different values of $t$ matched captions in each column. The results show that the best number of GIST matched captions is dataset dependent, however, GIST classification accuracy is not strongly affected by the number of GIST matched captions.

**Number of Captions.** We show the classification accuracy for different numbers of matched captions, $t$, on the 5-shot classification task for all datasets. Table 3 shows that the best number of matched captions is dataset dependent, however, GIST achieves strong classification accuracy independent of the number of matched captions per image. We hypothesize that datasets with lower image-text alignment in the pretrained CLIP representation space (approximated by the zero-shot accuracy) slightly benefits from more matched captions per image because the additional information helps with aligning text concepts to images.

**Caption Length.** Our results show that leveraging text descriptions improves image classification. However, there are many possible design choices for how to generate and leverage text descriptions. We find that automatically shortening the original GPT generated text and then using the shortened texts to fine-tune CLIP improves downstream fine-grained classification performance over using the original longer generated text or using only the label name in a template (e.g., the FLYP method). For 5-shot classification on Fitzpatrick40, the average accuracy and standard deviation are 42.18 (2.98) for FLYP, 42.25 (3.12) for our method using long GPT descriptions, and **44.10 (2.97)** for our method using the summarized GIST text descriptions.

**Captioning Model Comparison** We compare GIST to off-the-shelf captioning models, GIT and BLIP. For all generated captions, we append class label names. Using GIST captions to finetune achieves the best fine-grained classification on all shots for Fitzpatrick40 classification. The full-shot average accuracy and standard deviation is 74.17 (2.72) for GIT, 73.83 (2.75) for BLIP2, and **75.77 (2.67)** for GIST (ours). The 5-shot average accuracy and standard deviation is 42.96 (2.99) for GIT, 42.66 (3.03) for BLIP2, and **46.96 (3.13)** for GIST (ours).

## 5 CONCLUSION

We present GIST as a method for generating fine-grained image-specific text descriptions and demonstrate how to use GIST to learn an aligned image-text embedding space for improved *any-shot* classification. Compared to previous approaches, our text generation and image-text matching methods produce captions that are concise and specific to distinguishing image features. Our results demonstrate that fine-tuning CLIP, using contrastive learning, on our image-text pairs results in a well-aligned representation space, regardless of the original pretrained alignment. We show that applying GIST to image classification outperforms recent vision-language classification methods, black box probing methods, and image-only methods on full-shot and k-shot experiments for four diverse fine-grained datasets. One current limitation of GPT is that it does not always output accurate information, even with specific prompts. Thus, our current GIST method does require some limited manual checking for particular datasets (more detailed information in the Appendix.) We expect that as LLMs continue to improve, this will not be required.

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

# A    APPENDIX

## A.1    FITZPATRICK17K CLEANING

The original Fitzpatrick17k dataset (Groh et al., 2021) has 11,788 labeled images collected from two dermatology websites, DermAmin and Atlas Dermatologico. As mentioned in the CiRCLe paper (Pakzad et al., 2023), the Fitzpatrick17k dataset contains some erroneous images that are unrelated to dermatology, such as X-ray images (Figure 5 in the CiRCLe paper). Additionally, there is large class imbalance amongst the 114 class labels. We selected a subset of the dataset to manually clean and form a new dataset. To determine which classes to include in our new dataset, we organized the classes by number of examples. We selected classes that had under 100 images for ease of manual cleaning. We excluded six classes that had less than 100 images because they were diseases, such as neurotic excoriations, that require non-visual information to diagnose. Fitzpatrick40 has 2,609 images, which we split into 2,115 training, 222 validation, and 272 test. There are between 25 and 73 training images per class.

The labels in our Fitzpatrick40 dataset include: becker nevus, epidermal nevus, pilar cyst, erythema nodosum, stasis edema, sun damaged skin, xeroderma pigmentosum, behcets disease, perioral dermatitis, lentigo maligna, disseminated actinic porokeratosis, halo nevus, solid cystic basal cell carcinoma, port wine stain, livedo reticularis, lichen simplex, ichthyosis vulgaris, dyshidrotic eczema, congenital nevus, naevus comedonicus, aplasia cutis, porokeratosis of mibelli, calcinosis cutis, seborrheic keratosis, erythema elevatum diutinum, mucous cyst, pilomatricoma, erythema annulare centrifigum, acrodermatitis enteropathica, pustular psoriasis, pityriasis lichenoides chronica, nevocytic nevus, lichen amyloidosis, keratosis pilaris, granuloma pyogenic, epidermolysis bullosa, drug induced pigmentary changes, basal cell carcinoma morpheiform, acanthosis nigricans, nevus sebaceous of jadassohn.

### A.1.1 DUPLICATES AND NON-DERMATOLOGY IMAGES

After noticing there were duplicate images in the full dataset, we moved all duplicates and near duplicates in the test set to the training set. We found these duplicates by using the Sentence Transformers framework to embed the test set images into the CLIP embedding space (clip-ViT-B-32). We used cosine distance to find similar images. We found .95605 to be a good threshold to filter out duplicates in the test set. Once all duplicates were removed from the test set, we compared each remaining image to images in the validation and training set. We used the same threshold to filter out similar pairs and visually checked pairs with scores above the threshold. If we found a test set image that had near duplicates in the validation or training set, we moved both images to the training set. In addition to removing duplicates from the test set, we manually looked through the entire dataset and removed images irrelevant to dermatology (e.g. images similar to the ones mentioned in the CirCLe paper (Pakzad et al., 2023)).

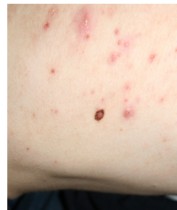 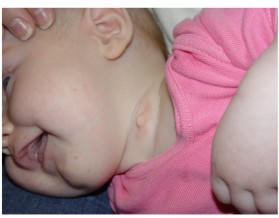 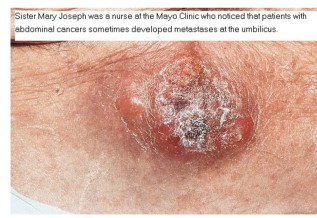

Eclipse Nevi
mislabeled as
Becker Nevus

Branchial Cleft Cyst
mislabeled as
Pilar Cyst

Sister Mary Joseph's Nodule
mislabeled as
Stasis Edema

Figure 5: Examples of mislabeled images in the Fitzpatrick17k dataset. We compared the dataset label with the ground truth labels on the original source dermatology website.

### A.1.2 INCORRECTLY LABELED IMAGES

The Fitzpatrick dataset was collected from two dermatology websites, DermaAmin and Atlas Dermatologico. We could not find ground truth labels for the Atlas Dermatologico images. However, when working with the 40 label dataset, we realized several DermaAmin images were labeled differently in the Fitzpatrick dataset than on the website (Figure 5). We manually checked all of the DermaAmin images in the 40 label set. We looked at the DermaAmin URL label and compared it to the fitzpatrick label. If the label was different, we used Google and ChatGPT (Brown et al., 2020) to check whether the labels were synonyms of each other. If the Fitzpatrick label described a different disease than the DermaAmin disease, then we either 1) removed the image if the ground truth label was not one of the 40 labels in our dataset, or 2) changed the image label if its ground truth label was one of the 40 labels different from its current Fitzpatrick dataset label. In some cases, the label synonyms were unclear (e.g. clinical websites differed in opinion). If we found a clinical website supporting that the labels were synonyms, we left the image as is. In future work, we would like to work with dermatologists to confirm the dataset cleaning choices.

For the training set, we removed a total of 223 images of the original 2338 training images and corrected the labels of 27 images. 223 mislabeled images had ground truth labels that were not one of the 40 labels, so we removed them entirely from the dataset. Seven images were labeled as mucous cysts when they were actually sebaceous cysts (pilar cysts). Twenty images were labeled as naevus comedonicus when the source website says they are congenital nevus.

For the validation set, we removed a total of 33 images of the original 255 validation images and corrected the labels of four images. 33 of the mislabeled images had ground truth labels that were not one of the 40 labels, so we removed them entirely from the dataset. One image was labeled as mucous cysts when they were actually sebaceous cysts (pilar cysts). Three images were labeled as naevus comedonicus when the source website says they are congenital nevus.

For the test set, we removed a total of 34 images of the 306 test images and corrected the labels of five images. 34 of the mislabeled images had ground truth labels that were not one of the 40 labels, so we removed them entirely from the dataset. Five images were labeled as naevus comedonicus when the source website says they are congenital nevus.

## A.2 CUB200 Mislabeled Images

The CUB200-2011 (Wah et al., 2011) is a well-cited, commonly-used dataset for image classification. The dataset offers great diversity of images and bird species. While working with the CUB200 dataset and examining validation images misclassified by our model, we noticed that some images in the dataset are mislabeled or are partially incorrect (e.g. have two different bird species present). We show a few of these images in Figure 6.

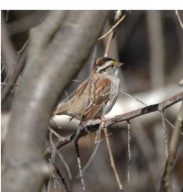 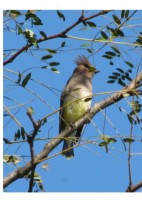 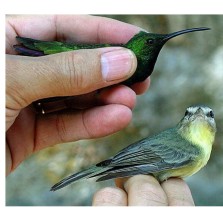 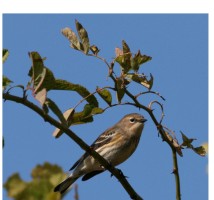

White-throated sparrow mislabeled as Grasshopper Sparrow

Cedar Waxwing mislabeled as Tennessee Warbler

Philadelphia Vireo (labeled) and Green-breasted Mango hummingbird present

Yellow-rumped Warbler mislabeled as Blue-headed Vireo

Figure 6: CUB200-2011 images that have incorrect or partially incorrect labels.

## A.3 GIST GPT Prompts

We provide the prompts we use for each dataset to generate the long GPT captions.

Given a set of generic body parts: face, neck, arms, torso, legs, torso, scalp, hands, feet. For the Fitzpatrick40 labels, we generate long class captions with:

```
"You are a dermatology disease describer. Describe what an image of
<<disease>> might look like on a person's <<body part>>."
```

For the CUB200-2011 labels, we generate long class captions for the male and female bird genders with:

```
"You are a bird species describer. Describe what an image of a <<gender>>
<<species>> might look like."
```

For the Flowers102 labels, we generate long class captions for flower species with:

```
"You are a flower describer. Describe what an image of a flower of
<<species>> might look like."
```

For the FGVC labels, we generate long class captions for aircraft models with:

```
"You are an airplane model describer. Please describe distinguishing
characteristics of what the plane looks like in 2-3 sentences. What
would a plane of type <<model>> look like?"
```

## A.4 GIST Caption Length

Our method uses GPT to generate long class-specific descriptions. After matching each training image to a description, our method uses GPT again to summarize the matched descriptions into concise captions. We show examples of the original long and the summarized short captions in Table 4. As explained in the main paper and supported by our method analysis studies, having longer captions is useful for the image-caption matching phase. Including details, such as body part, in the longer captions increases the likelihood that matched long captions will accurately and specifically describe the images. After this matching, we find that using GPT to summarize the captions results in descriptions that contain the key visual features needed for classification.

| Class Labels | Dataset | Long Caption | Short Caption |
|---|---|---|---|
| Levido Reticularis | Fitzpatrick40 | "The image shows a person's back. The skin is mottled with a network of red or purple discoloration. The discoloration appears in a reticular pattern with a lace-like structure." | "mottled red or purple discoloration in reticular pattern" |
| Erythema Nodosum | Fitzpatrick40 | "The image shows a person's leg with raised, red, tender lumps. The lumps are usually on the shins, and may be painful when touched. The lumps are usually red to purple in color, and may be swollen." | "raised, red, tender lumps" |
| Red-winged Blackbird | CUB200 | "The male Red-winged Blackbird has a black body with a red and yellow wing patch on each shoulder. It has a sharp, pointed black beak and a sleek in appearance" | "black body with red and yellow wing patch, sharp pointed black beak, sleek in appearance." |
| Yellow-billed Cuckoo | CUB200 | "The female Yellow-billed Cuckoo has a slender, sleek body with a long tail. She has a brownish-grey back, a white chest and belly, and prominent white spots on her dark tail. The bill is yellow on the lower mandible and black on the upper mandible." | "brownish-grey back, white chest and belly, white spots on tail, yellow lower mandible, black upper mandible." |
| Frangipani | Flowers102 | "Frangipani flowers have large, showy blooms with five broad, waxy petals that create a visually captivating display. The petals can appear in various colors, including shades of white, yellow, pink, or orange with a yellow or white center. The petals often have a smooth texture." | "large, waxy, and fragrant flowers that have five to nine petals in a star shape and come in a variety of colors such as white, pink, yellow, and red" |
| Poinsettia | Flowers102 | "The poinsettia flower is captivating with its vibrant colors and unique structure. The bracts, which are modified leaves, are the main attraction of this flower. They are typically in shades of red, but can also be found in pink, white, or even specks of yellow. The bracts are large and oval-shaped, with a slightly wavy or scalloped edge that adds a touch of elegance." | "deep red, star-shaped flowers surrounded by bright green, leaf-like bracts" |
| 727-200 | FGVC Aircraft | "The Boeing 727-200 is a trijet, narrow-body airliner known for its distinctive T-tail configuration. It has a relatively short and stout fuselage with three engines mounted at the rear tail section. The aircraft's wings are swept-back and positioned low on the fuselage. Its unmistakable appearance is further enhanced by the distinctive trijet engine arrangement and the characteristic dorsal intake on top of the tail." | "trijet, narrow-body airliner, T-tail configuration, short and stout fuselage, three rear-mounted engines, swept-back low wings, distinctive trijet engine arrangement, dorsal intake on top of tail" |
| 747-100 | FGVC Aircraft | "The Boeing 747-100 is a large wide-body jetliner renowned for its iconic appearance. It features a distinctive hump-shaped upper deck, creating an instantly recognizable profile. With four engines mounted under the wings, the 747-100 has a long and elegant fuselage." | "large, wide-body jetliner, hump-shaped upper deck, four engines mounted under wings, long and elegant fuselage" |

Table 4: The original long captions generated by GPT are useful for pairing images to captions. The summarized GPT captions provide more concise descriptions of important visual features for classification.

| Image | GIT | BLIP2 | GIST |
|---|---|---|---|
|  | "a man with a bandaged foot" | "a person with a foot with a rash on it" | "red and white patches with raised, inflamed bumps and flaky scales" |
|  | "a close up of a child's face with food on it" | "a baby with a toothbrush in its mouth" | "yellowish-brown patch of discoloration" |
|  | "the inside of the ear" | "a man with a large ear infection" | "flaky, white-gray patches of skin with redness and itching" |

Figure 7: Example GIT, BLIP2 and GIST generated captions for Fitzpatrick40 images. For the dermatology images, GIST contains the most disease relevant information in comparison to off-the-shelf captioning models.

## A.5 OFF-THE-SHELF CAPTIONING COMPARISONS

Figure 7 shows GIST captioning qualitative results in comparison to off-the-shelf captioning methods for three images from the Fitzpatrick40 dataset. Clearly, the GIST captions contain more disease specific captions. Some of the off-the-shelf captions are incorrect. For example, in the second row, BLIP2 identifies a toothbrush in the mouth of the baby even though neither the baby's mouth nor a toothbrush are shown.

| Experiments | GIT | Ours |
|---|---|---|
| Full | 64.83 (2.98) | **74.59 (2.62)** |
| 5-shot *Top-1* | 24.02 (1.21) | **44.10 (2.97)** |
| 3-shot *Top-1* | 12.13 (2.40) | **40.24 (2.95)** |
| 1-shot *Top-1* | 1.42 (0.29) | **19.10 (2.19)** |

Table 5: Fine-tuning CLIP on GIST captions outperforms fine-tuning an image captioning model on GIST for classification on the Fitzpatrick40 dataset. We fine-tune GIT to output a caption containing the class prediction. We report the average accuracy and standard deviation for full-shot and few-shot classification. For the full-shot, we compute statistics over 1000 bootstrapped samples of the test set. For few-shot classification, we compute statistics over three k-shot samples. The results show that for the same GIST captions, fine-tuning CLIP results in better classification accuracy than fine-tuning the GIT captioning method.

## A.6 VISUAL GROUNDING COMPARISON

We compare our method to a visual grounding image captioning method, GIT (Wang et al., 2022), for both fine-grained text generation and fine-grained image classification. As shown in the bottom row of Figure 8, the pretrained GIT model outputs coarse captions that mostly focus on object-level details. In contrast, our GIST captions capture more fine-grained details.

We additionally compare our fine-tuning CLIP approach with fine-tuning GIT for classification using the GIST fine-grained image-text pairs. We fine-tune GIT on our paired images and GPT-generated text descriptions with three captions per image. We append the ground truth class name to the text descriptions, in the format "description : class name", to form the image training captions. At inference time, we classify an image by comparing the class name in the caption to the ground

truth class name. Table 5 shows the average accuracy and standard deviation for the full-shot and k-shot experiments on the Fitzpatrick40 dataset. Our fine-tuning with the GPT-generated matched captions outperforms fine-tuning GIT with the same image-text pairs. Figure 8 shows the GIT generated captions for two test images across different classification experiments. Without fine-tuning, GIT focuses on objects or coarse-grained details without domain knowledge. When fine-tuning on the full training set, GIT learns to properly describe the important image features and predict the correct class name in many cases. In the *k*-shot experiments, however, GIT quickly overfits to the training examples. The descriptions contain useful features, but are attributed to the wrong class.

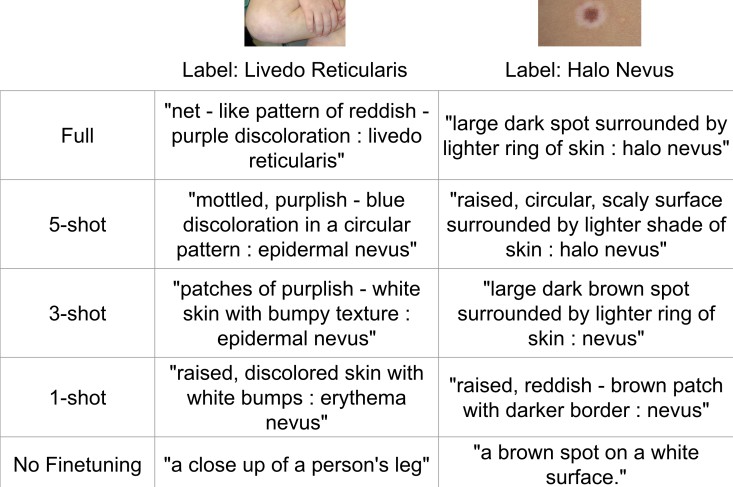

| | Label: Livedo Reticularis | Label: Halo Nevus |
|---|---|---|
| Full | "net - like pattern of reddish - purple discoloration : livedo reticularis" | "large dark spot surrounded by lighter ring of skin : halo nevus" |
| 5-shot | "mottled, purplish - blue discoloration in a circular pattern : epidermal nevus" | "raised, circular, scaly surface surrounded by lighter shade of skin : halo nevus" |
| 3-shot | "patches of purplish - white skin with bumpy texture : epidermal nevus" | "large dark brown spot surrounded by lighter ring of skin : nevus" |
| 1-shot | "raised, discolored skin with white bumps : erythema nevus" | "raised, reddish - brown patch with darker border : nevus" |
| No Finetuning | "a close up of a person's leg" | "a brown spot on a white surface." |

Figure 8: Examples of generated captions and class predictions from GIT for two Fitzpatrick40 test images. While the captions provide useful information and accurate predictions for the full-shot experiment, GIT overfits on the few-shot experiments. Without fine-tuning, GIT lacks fine-grained domain details.

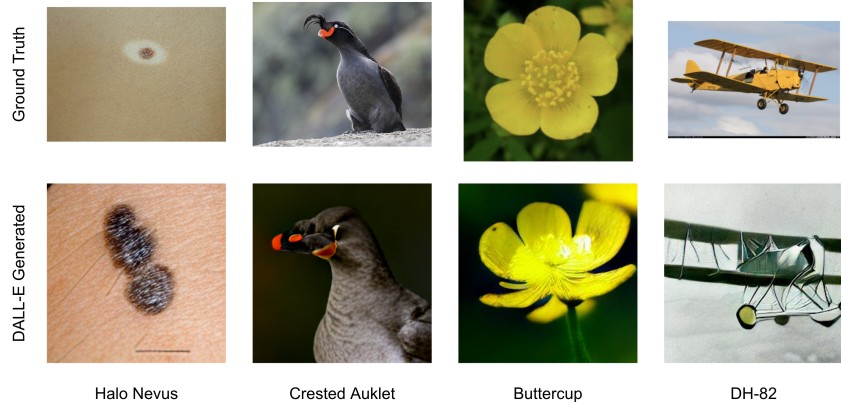

Halo Nevus          Crested Auklet          Buttercup          DH-82

Figure 9: Examples of ground truth images from our dataset compared to the DALL-E images generated for the CaFo method. For the Fitzpatrick40 dataset, DALL-E is unable to generate realistic images for most diseases. For example, the generated halo nevus image is missing the key light-colored halo feature. For the other three datasets, DALL-E is able to generate coarse details, such as color or shape, correctly, but the images don't always appear realistic.

## A.7 ADDITIONAL QUALITATIVE RESULTS COMPARING VISION-LANGUAGE METHODS

### A.7.1 DALL-E IMAGES

Figure 9 shows DALL-E generated images from each dataset and their ground truth labels, which are used for the CaFo baseline method. While DALL-E is able to generate correct coarse-grain and fine-

| | Training Images | CaFo Generated Text | LaBo Candidate Concepts | FLYP Text | GIST Matched Text (Ours) |
|---|---|---|---|---|---|
| Fitzpatrick40 | | "Erythema nodosum can be identified by the presence of tender, red, or purple nodules on the front of the legs, thighs, or ankles. Other symptoms include fever, joint pain, drainage from lesions, and swelling in the legs." | "and tender bumps" "bright red bumps" "clustered together" | "A photo of erythema nodosum" | 'erythema nodosum: raised, red lesions with bruise-like appearance and tender skin' |
| CUB200 | | "The Bobolink is a small bird, measuring about 7 inches in length. Males are distinctive: in breeding plumage they have a jet-black body and a white back and rump. The back of the male's head is yellow. Females and non-breeding males have a mostly brown, streaky coloration with a paler chest, similar to a sparrow" | "black and white face" "white belly and undertail" "black stripes on its back" | "A photo of a bobolink" | "bobolink: black body, white back, bright yellow nape, off-white underparts, bold white markings on wings" |
| Flowers102 | | "A photo of a flower silverbush, it has a lavendar color or sometimes white color flower with tiny petals. The center of the flower is often yellow color. The leaves are oblong." | "small, delicate flower with petals that are a pale silver color" "and a small yellow" "used in herbal teas and remedies" | "A photo of a silverbush" | "silverbush: small, white, star-shaped flowers with five petals and bright yellow centers" |
| FGVC | | "The Yak-42 is a Russian-made aircraft that can seat up to 140 passengers. It has a triangular shape with three engines, and a wide, sloping nose. The Yak-42 first flew in 1982 and is still in production" | "blue and have red stripes" "twin-engine turboprop aircraft designed in the soviet union" "entered service with aeroflot in 1984" | "A photo of a Yak-42" | "Yak-42: medium-range, three-engine jet airliner, low-wing design with rear engine pods, T-tail configuration, sleek and streamlined fuselage" |

Figure 10: Generated captions and concepts from each method. Our method, GIST, generates concise but detailed captions. The baselines either generate short captions that lack visual features or generate long captions that have superfluous words.

grain details for flowers, birds and aircrafts, the photos are not fully realistic and lack sharp details. Furthermore, it often misses important features for the dermatology dataset. For example, the halo nevus image is missing the distinctive lighter pigmented halo around the dark mole. Our results in Section A.8 show that the DALL-E images help the CaFo method in the 1-shot classification setup, but do not boost performance compared to our method for other k-shot experiments. We hypothesize this is because the DALL-E images convey enough visual features to add information in the 1-shot setup, but lack enough realism and important features to help when there is more than one ground truth image per class.

### A.7.2 GENERATED TEXT

We show additional examples of the generated text from each baseline method and our method in Figure 10.

### A.7.3 LEARNED REPRESENTATION FEATURES

We also show an example of learned features from our fine-tuned representation space and CLIP's pretrained representation space. Figure 11 shows a test image (left column) that our method classifies correctly and CLIP LP classifies incorrectly when trained on the full training set. The remaining columns show the nearest neighbor training images to the test image using cosine similarity over the L2 normalized embeddings. For the GIST method, we show the top matched caption for each training nearest neighbor image. For CLIP, we show the ground truth label for each image. In this dermatology example shown in Figure 11, GIST and CLIP have most of the same nearest neighbors. However, GIST corrects the fourth nearest neighbor compared to CLIP. Even though the GIST nearest neighbor in the fifth column does not visually look similar to the test image, they are close in representation because they share class-specific features, as demonstrated in the text description (e.g. "circular patch of baldness"). The GIST fine-tuned method seems to capture domain-specific features, whereas CLIP focuses on image-specific features. The GIST representation is better for classification because the representation space should capture class-specific, rather than image-specific, features.

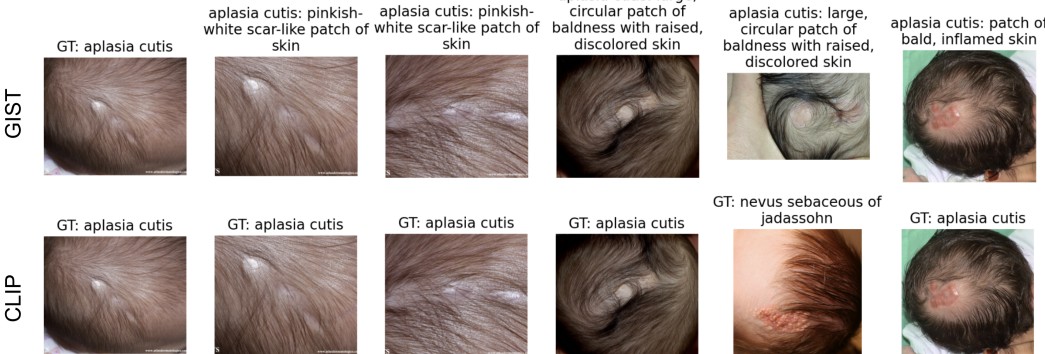

Figure 11: A qualitative example of the GIST fine-tuned representation space and the pretrained CLIP representation space for a Fitzpatrick40 test example that our method predicts correctly and CLIP LP predicts incorrectly. The left column shows a test image with its ground truth (GT) label. The other five columns show the five nearest neighbor training images, in descending order, of the test image with either the corresponding top matched caption (GIST) or the ground truth class name (CLIP). Our approach captures domain-specific details (e.g. "circular patch of baldness"), whereas CLIP focuses on image features that may not capture class differences.

| Methods | Fitzpatrick40 *Top-3* | CUB200 *Top-3* | Flowers102 *Top-3* | FGVC *Top-3* |
|---|---|---|---|---|
| ResNet | 85.67 (2.12) | 92.63 (0.35) | 99.88 (0.07) | 89.11 (0.79) |
| ResNet-RW | 85.62 (2.16) | 92.35 (0.35) | 89.11 (0.79) | 88.98 (0.53) |
| CLIP ZS | 7.40 (1.61) | 84.47 (0.47) | 85.45 (0.70) | 61.12 (0.85) |
| CLIP LP | 90.13 (1.76) | 96.32 (0.25) | **99.92 (0.06)** | 86.04 (0.60) |
| LaBo | 81.62 (0.20) | 95.17 (0.01) | 99.88 (0.05) | 85.27 (0.17) |
| FLYP | 89.73 (1.89) | 96.58 (0.28) | **99.92 (0.06)** | 90.29 (0.50) |
| **GIST (Ours)** | **90.81 (1.78)** | **96.86 (0.26)** | **99.92 (0.06)** | **90.42 (0.51)** |

Table 6: Fine-tuning with GIST captions achieves the best top-3 accuracy for full-shot classification across all datasets. We report the average accuracy and standard deviation over 1000 bootstrapped samples of the test set.

## A.8 Additional Quantitative Results Comparing Vision-Language Methods

We show top-3 accuracy for full-shot classification in Table 6. Our method outperforms the other methods on three of the four datasets and ties with FLYP and CLIP LP on the Flowers102 dataset. While we overlap for top-3 accuracy on Flowers102, we show in the main paper that we outperform all methods for all datasets with top-1 accuracy.

We show top-1 and top-3 accuracy for $k$-shot experiments, where $k$=1,3,5, in Table 7, Table 8, Table 9, and Table 10 for the Fitzpatrick40, CUB200, Flowers102, and FGVC-Aircraft datasets, respectively. Our method outperforms the method on top-1 accuracy for both 5-shot and 3-shot classification. Additionally, we outperform other methods on top-3 accuracy for 5-shot classification. CaFo performs best for the 1-shot setting. CaFo generates DALL-E (Ramesh et al., 2021) images of each class to aid in few-shot classification. These DALL-E images are relatively, but not perfectly accurate, as shown in Figure 4 and Figure 9. We hypothesize that the relative improvement of CaFo over GIST is due to GIST overfitting in the 1-shot case and that the generated images, while imperfect, are helpful in the very-low data regime.

| Methods | 5-shot | | 3-shot | | 1-shot | |
|---|---|---|---|---|---|---|
| | *Top-1* | *Top-3* | *Top-1* | *Top-3* | *Top-1* | *Top-3* |
| ResNet | 31.62 (3.69) | 50.37 (2.61) | 19.98 (1.73) | 40.32 (1.35) | 12.13 (1.97) | 25.73 (0.79) |
| CLIP ZS | 2.94 (1.06) | 7.40 (1.61) | 2.94 (1.06) | 7.40 (1.61) | 2.94 (1.06) | 7.40 (1.61) |
| CLIP LP | 41.49 (2.99) | 63.59 (3.01) | 36.07 (2.94) | 58.81 (2.99) | 18.77 (2.23) | 37.05 (2.88) |
| LaBo | 32.17 (0.25) | 60.47 (0.26) | 31.62 (0.08) | 54.78 (0.12) | 24.63 (1.04) | **53.68 (0.58)** |
| CaFo | 40.32 (0.63) | 65.93 (1.83) | 31.62 (2.67) | 57.12 (2.11) | **25.37 (1.08)** | 47.92 (1.21) |
| FLYP | 42.18 (2.98) | 65.50 (2.96) | 39.87 (3.00) | 61.20 (2.97) | 19.90 (2.37) | 38.35 (2.95) |
| **GIST (Ours)** | **46.96 (3.13)** | **71.29 (2.82)** | **41.24 (2.95)** | **63.60 (3.01)** | 21.06 (2.38) | 38.28 (2.91) |

Table 7: Fine-tuning with GIST captions achieves the best top-1 and top-3 accuracy on Fitzpatrick40 for the 5-shot and 3-shot regimes. We report the average accuracy and standard deviation over three k-shot samples. For the CLIP zero-shot experiments, we compute statistics over 1000 bootstrapped samples of the test set.

| Methods | 5-shot | | 3-shot | | 1-shot | |
|---|---|---|---|---|---|---|
| | *Top-1* | *Top-3* | *Top-1* | *Top-3* | *Top-1* | *Top-3* |
| ResNet | 49.83 (0.47) | 72.05 (0.24) | 34.89 (0.55) | 55.04 (0.92) | 13.37 (0.87) | 24.65 (1.16) |
| CLIP ZS | 63.29 (0.63) | 84.47 (0.47) | 63.29 (0.63) | 84.47 (0.47) | 63.29 (0.63) | 84.47 (0.47) |
| CLIP LP | 76.41 (0.57) | 91.97 (0.36) | 70.42 (0.59) | 88.72 (0.42) | 46.99 (0.68) | 69.20 (0.62) |
| LaBo | 73.66 (0.07) | 90.51 (0.10) | 69.15 (0.04) | 87.27 (0.06) | 54.21 (0.06) | 76.79 (0.08) |
| CaFo | 69.48 (0.28) | 88.99 (0.24) | 65.52 (1.21) | 86.41 (2.17) | **66.48 (0.38)** | **88.29 (0.08)** |
| FLYP | 78.68 (0.52) | 93.03 (0.32) | 73.11 (0.57) | 90.47 (0.38) | 50.28 (0.69) | 70.68 (0.62) |
| **GIST (Ours)** | **79.52 (0.55)** | **93.29 (0.34)** | **74.30 (0.57)** | **90.88 (0.38)** | 51.50 (0.66) | 72.88 (0.60) |

Table 8: Fine-tuning with GIST captions achieves the best top-1 and top-3 accuracy on CUB200-2011 for the 5-shot and 3-shot regimes. We report the average accuracy and standard deviation over three k-shot samples. For the CLIP zero-shot experiments, we compute statistics over 1000 bootstrapped samples of the test set.

| Methods | 5-shot | | 3-shot | | 1-shot | |
|---|---|---|---|---|---|---|
| | *Top-1* | *Top-3* | *Top-1* | *Top-3* | *Top-1* | *Top-3* |
| ResNet | 81.35 (0.35) | 91.30 (0.30) | 72.84 (1.31) | 86.45 (0.47) | 48.47 (0.61) | 64.78 (0.28) |
| CLIP ZS | 76.64 (0.84) | 85.45 (0.70) | 76.64 (0.84) | 85.45 (0.70) | 76.64 (0.84) | 85.45 (0.70) |
| CLIP LP | 97.48 (0.32) | 99.72 (0.11) | 96.28 (0.38) | 99.63 (0.12) | 82.93 (0.73) | 94.50 (0.45) |
| LaBo | 96.06 (0.02) | 99.11 (0.01) | 92.75 (0.04) | 97.28 (0.23) | 80.63 (0.05) | 93.24 (0.14) |
| CaFo | 95.40 (0.68) | 99.35 (0.15) | 93.42 (0.56) | 98.89 (0.23) | **86.50 (0.95)** | **95.98 (0.23)** |
| FLYP | 97.80 (0.33) | 99.76 (0.10) | 96.89 (0.33) | **99.76 (0.10)** | 84.56 (0.71) | 95.03 (0.43) |
| **GIST (Ours)** | **97.88 (0.29)** | **99.80 (0.09)** | **97.02 (0.34)** | 99.72 (0.10) | 85.63 (0.68) | 95.10 (0.43) |

Table 9: Fine-tuning with GIST captions achieves the best top-1 and top-3 accuracy on Flowers-102 for the 5-shot and comparable accuracy for 3-shot. We report the average accuracy and standard deviation over three k-shot samples. For the CLIP zero-shot experiments, we compute statistics over 1000 bootstrapped samples of the test set.

| Methods | 5-shot | | 3-shot | | 1-shot | |
|---|---|---|---|---|---|---|
| | *Top-1* | *Top-3* | *Top-1* | *Top-3* | *Top-1* | *Top-3* |
| ResNet | 22.31 (0.54) | 39.27 (1.34) | 14.22 (0.37) | 26.94 (0.84) | 7.02 (0.56) | 13.83 (0.70) |
| CLIP ZS | 32.18 (0.83) | 61.12 (0.85) | 32.18 (0.83) | 61.12 (0.85) | 32.18 (0.83) | 61.12 (0.85) |
| CLIP LP | 45.13 (0.86) | 68.79 (0.77) | 39.51 (0.87) | 62.76 (0.86) | 26.01 (0.78) | 44.82 (0.85) |
| LaBo | 45.20 (0.02) | 70.33 (0.04) | 40.52 (0.03) | 65.39 (0.15) | 32.58 (0.28) | 56.35 (0.42) |
| CaFo | 43.16 (1.15) | 70.73 (0.97) | 40.85 (1.46) | **68.44 (1.38)** | **40.05 (0.55)** | **69.78** (0.17) |
| FLYP | 48.07 (0.87) | 70.00 (0.79) | 39.45 (0.86) | 63.28 (0.86) | 27.09 (0.78) | 46.87 (0.87) |
| **GIST (Ours)** | **49.44 (0.86)** | **72.93 (0.78)** | **41.84 (0.88)** | 67.30 (0.80) | 28.52 (0.78) | 48.85 (0.87) |

Table 10: Fine-tuning with GIST captions achieves the best top-1 and accuracy on FGVC Aircraft for the 5-shot and comparable accuracy for 3-shot. We report the average accuracy and standard deviation over three k-shot samples. For the CLIP zero-shot experiments, we compute statistics over 1000 bootstrapped samples of the test set.

| **Methods** | RN50 | ViT-B/16 | ViT-B/32 | ViT-L/14 | ViT-L/14@336px |
|---|---|---|---|---|---|
| CLIP LP | 31.97 (2.85) | 38.37 (2.93) | 35.88 (2.90) | 42.27 (3.14) | 41.49 (2.99) |
| CLIP ZS | 1.48 (0.73) | 2.98 (1.03) | 1.79 (0.81) | 1.48 (0.74) | 2.94 (1.06) |
| LaBo | 23.16 (0.52) | 31.44 (0.26) | 29.60 (0.26) | 33.91 (0.26) | 32.17 (0.25) |
| CaFo | 32.48 (4.40) | 37.01 (3.04) | 35.91 (2.00) | 40.32 (1.71) | 40.32 (0.63) |
| FLYP | 32.58 (2.84) | 40.35 (3.03) | 37.25 (2.87) | 43.97 (3.05) | 42.18 (2.98) |
| **GIST (Ours)** | **33.40 (2.81)** | **43.91 (3.01)** | **42.08 (2.97)** | **47.45 (3.12)** | **46.96 (3.13)** |

Table 11: GIST-generated captions improves on 5-shot Fitzpatrick40 classification over a selection of different CLIP models. The top-3 GIST matched captions are used for fine-tuning. The relative improvement of the GIST-generated captions over the baselines depends on the CLIP model used for fine-tuning.

## A.9 CLASSIFICATION RESULTS FOR DIFFERENT CLIP MODELS

In Table 11, we show the full results for comparing Fitzpatrick40 5-shot classification accuracy across different CLIP methods. While classification accuracy can vary depending on the CLIP model, our method outperforms other vision-language classification methods regardless of the CLIP model used.

## B LIMITATIONS

One current limitation of GPT is that it does not always output accurate information, even with specific prompts. For our GIST approach, we found GPT-3 to work well without modification for two datasets, Fitzpatrick40 and Flowers102. For FGVC-Aircraft, the GPT-3 captions required manual checking of the captions for rare inaccurate information for particular classes. Most captions did not require correction and the process took approximately 20 minutes of manual work. For the CUB200 dataset, the majority of GPT-3 captions were incorrect. We therefore switched to using GPT-4 for the CUB200 captions. These captions still required manual checking and this process took less than a day of manual work. There is a slight improvement in accuracy when the captions are manually checked and cleaned, however, even with the raw GPT-4 captions (no manual cleaning), our method still has a higher average accuracy than the baselines. For example, the CUB200 5-shot classification top-1 accuracy for fine-tuning on the cleaned GIST captions is 79.52 (0.55) versus 79.22 (0.51) when fine-tuning on the raw GPT-4 captions. The manual checking is currently a one-time upfront cost for GIST. As LLMs improve, this will not be required.

