# OpenReview forum: "GIST: Generating Image-Specific Text for Fine-grained Object Representations"
_ICLR.cc/2024/Conference — Submitted to ICLR 2024_

### Official Review · Reviewer_om6n · 2023-10-31

**Soundness:** 3 good
**Presentation:** 3 good
**Contribution:** 3 good
**Rating:** 5
**Confidence:** 4

**Summary:**

This paper proposes a method to enhance VL model ability for few-shot classification. Using the names of interested classes, LLM generates diverse and descriptive explanations of each class. The CLIP model matches each image in the dataset with the generated explanations, resulting in an image-synthetic caption dataset. Using the dataset, the CLIP model can discriminate the fine-grained information better. The proposed method achieves better results than comparable methods.

**Strengths:**

+ The proposed method is simple, straightforward, and sound.

+ The paper is clearly written and easy to understand.

+ The proposed method achieves better results than comparable methods.

**Weaknesses:**

- The proposed method requires domain (dataset) specific designs such as prompts for LLM. However, in a few-shot setting, it is hard to determine the designs because of the scarcity of validation and test sets. This makes the proposed method less general to real-world few-shot tasks.

- The paper lacks of comparison with other fine-tuning techniques such as prompt tuning approaches. Please analyze and compare with other families of VL finetuning.

- If I understood correctly, the proposed method requires full images for the target domain. In a real-world setting, it may be hard to obtain such images, especially for medical images.

- Why is a summary is required? We can see the performance improvement by doing this, but I found no proper explanations for why.

- Manual cleaning mentioned in Section 5 seems not fancy. In a specific field, manual cleaning requires an expensive effort of experts such as the medical field. In addition, instead of manually cleaning, we can make labels for more images under the same effort, which is likely to produce much more improvements than manual cleaning.

**Questions:**

- Instead of the process of generating captions using LLM and matching with CLIP, how if the trained vision-llm models (e.g., BLIP-2, Flamingo) are used to generate captions?

- The generated explanations are exclusively matched to images? Or the same captions can be matched to multiple images? If so, please statistics how frequently each explanation is matched to the images.

---

> ### Author Response · Authors · 2023-11-12
>
> We thank the reviewer for their careful review and feedback. We appreciate that the reviewer recognizes the improvement of GIST over comparable methods. We address the weaknesses and questions raised as follows:
>
> Weaknesses:
>
> 1. We don’t require prompt tuning or in-depth domain knowledge for the LLM prompts. Our prompts are listed in Appendix section A.3. Our prompting approach is identical for all-shot settings. We add material to the main text about prompting to clarify this.
>
> 2. We did provide experimental results for VL finetuning using image captioning models and also compare against two CVPR 2023 VL finetuning papers. Did the reviewer have a specific method in mind?
>
> 3. Could the reviewer please expand on what is meant by “full images”? GIST assumes access to an image dataset with class labels but has no restrictions on the number of images per class. The full pipeline can be applied to domains that have very few images.
>
> 4. Text summarization is not required, but does improve results (Section 4.5 of submission). Our method using the longer, original text outperforms the baselines. However, we achieve even better results when we summarize the captions. The intuition behind this is that the summarized captions remove extraneous words and are more similar to the format of the original CLIP training data. We add this discussion in the revision and thank the reviewer for the suggestion.
>
> 5. We show that even WITHOUT manual cleaning, our method can outperform the baselines. As LLMs improve, the manual cleaning step will become far less useful.
>
> Questions:
>
> 1. This is an excellent question. We provide experimental results for using BLIP-2 and GIT to generate captions in Section 4.5: Captioning Model Comparison. The full- shot average accuracy and standard deviation is 74.17 (2.72) for GIT, 73.83 (2.75) for BLIP-2, and 75.77 (2.67) for GIST (ours). The 5-shot average accuracy and standard deviation is 42.96 (2.99) for GIT, 42.66 (3.03) for BLIP-2, and 46.96 (3.13) for GIST (ours). We show additional comparison results in the Appendix in Figure 7, Table 5, and Sections A.5 and A.6. Our method outperforms all of these other approaches. Would the reviewer suggest that we move some of the results from the Appendix into the main paper? If so, is there material that the reviewer thinks we could move from the main body into the Appendix?
>
> 2. The same caption can be matched to multiple images since the same fine-grained features can appear in many images. The average number of images matched per captions varies per dataset. For the full Fitzpatrick40 dataset, for example, each caption is matched to an average of 2.8 images. For all five-shot datasets, each matched caption is matched to an average of 1.3 images. Could the reviewer please additionally clarify which other statistics they would like?

---

### Official Review · Reviewer_aPA7 · 2023-10-31

**Soundness:** 3 good
**Presentation:** 3 good
**Contribution:** 2 fair
**Rating:** 5
**Confidence:** 5

**Summary:**

This paper presents a method of generating image-specific text by prompting LLM for fine-grained object representations. In particular, the authors provide a three-step workflow, i.e., 1) prompting GPT-3 with domain-specific prompts to generate detailed  candidate text descriptions; 2) using CLIP to match each training each to the candidate text set; 3) summarizing the matched text via LLM to construct image-text pairs for fine-tuning CLIP image encoder. In addition, the learned representation is useful for fine-grained image classification. Besides, the authors provide a new fine-grained image classification dataset, Fitzpatrick40. Experimental results proved the effectiveness of the proposed method.

**Strengths:**

(1) The motivation of prompting LLM for fine-grained classification is well presented.

(2) The explanations and illustrations of the three-step workflow is well-formulated and mostly clear.

**Weaknesses:**

(1) The motivation of prompting LLM for visual classification is not that novel to the community and the authors listed (Maniparambil et al., 2023) as an example.

(2) The contribution is limited since the core idea is the same as typical data labeling workflow that uses LLM. More specifically, during prompting LLM, the user-provided prompt is not coming for free, it is also a kind of human knowledge or preference prior. There is no discussion on this difficulty of prompt preparation as compared to typical easy prompt such as “a photo of class name”. Thus, the workflow of generating image-text pairs has no difference from data labeling workflow that uses LLM.

(3) In the experiments, Table 1, the proposed workflow which uses GPT-3, CLIP and intricate prompts, does not get significant improvement over FLYP which uses CLIP and a typical easy prompt, though the comparison setting is unfair.

(4) The most conflicting method (Maniparambil et al., 2023) is not compared in the experiments.

**Questions:**

No.

---

> ### Author Response · Authors · 2023-11-12
>
> We thank the reviewer for their careful review and feedback. We address the individual weaknesses raised below:
>
> Weaknesses:
>
> 1, 4. We agree that prompting LLMs for visual classification is not novel. However, we believe that our approach for doing this is novel, and we show that it outperforms previous methods. Our way of matching images to LLM-generated captions is a novel part of our method that has not been used in previous methods. We show that our method outperforms three CVPR 2023 methods that use text for visual classification.
> We cited Maniparambil et. al. 2023 as concurrent work since the workshop proceedings were released shortly before the ICLR 2023 deadline. We will discuss it in more detail in the revision.  Maniparambil aims to improve accuracy for zero-shot CLIP. In contrast, our generated image-text pairs allow us to fine tune CLIP to achieve higher overall classification accuracy.
>
> 2. We provide our text prompts in Appendix section A.3 and show that they do not require in-depth domain knowledge. We should have included some of this material in the main text. We do not require prompt tuning and only use one prompt template per domain. The user-provided prompt does not require a significant human load and the prompt can be used across many classes and datasets within the domain. We show in Figure 4 and Figure 10 that our prompting method results in more detailed and relevant text descriptions than the prompting methods used in LaBo and CaFo.
>
> 3. Our improvements over FLYP are statistically significant, using a paired t-test. They are also meaningful. For the majority of full-shot and five-shot evaluation settings in Table 1, the absolute accuracy improvement of GIST over FLYP is greater than the absolute improvement of FLYP over off-the-shelf CLIP linear probing (CLIP LP).

---

### Official Review · Reviewer_zGRf · 2023-11-01

**Soundness:** 2 fair
**Presentation:** 2 fair
**Contribution:** 2 fair
**Rating:** 6
**Confidence:** 4

**Summary:**

This paper presents a new method for generating and matching fine-grained descriptions for images, which is named GIST. The proposed method GIST adopts a large language model to generate fine-grained texts with carefully designed prompts and then uses pre-trained CLIP to match the images with generated texts.
Based on the generated image-text pairs, this paper adopts a pre-trained CLIP and trains a classifier for downstream tasks. The experiments are conducted in both full-shot and few-shot settings. The experimental results show that the proposed GIST outperforms previous approaches and the baseline CLIP.

**Strengths:**

1. This paper presents a new method named GIST for fine-grained classification.
2. The proposed method adopts a large language model to generate domain-specific class descriptions and matches the texts and images with a pre-trained vision-language model.
3. This paper trains a CLIP classifier with the generated and matched descriptions.
4. The proposed GIST achieves good results on several benchmarks with full/few-shot settings.

**Weaknesses:**

1. I'm concerned about the technical contribution of this paper. Using a large language model to generate/augment text for CLIP training has been explored in several works [1,2].
2. This paper lacks the ablations about matching texts and images. It's unclear whether the matching based on a pre-trained CLIP will impact the downstream tasks. In addition, I'm concerned about how many captions are matched to one image and whether more captions will help.
3. This paper lacks studies on the impact of fine-tuning CLIP and whether more data (images) will further improve.

[1] Fan et.al. Improving CLIP Training with Language Rewrites. NeurIPS 2023.
[2] Maniparambil et.al. Enhancing CLIP with GPT-4: Harnessing Visual Descriptions as Prompts. ICCVW 2023.

**Questions:**

1. What are text inputs for the proposed GIST, the standard text prompts with labels or the text descriptions?
2. I'm concerned about the performance of fine-tuning CLIP with both short text descriptions and the original texts, and how about fine-tuning the CLIP with longer texts while fine-tuning classifiers with shorter descriptions.

---

> ### Author Response · Authors · 2023-11-12
>
> We thank the reviewer for their careful review and feedback. We appreciate that the reviewer recognized that GIST achieves good results across many evaluation settings and appreciate the reviewer’s suggestions to improve the paper. We address the individual weaknesses and questions raised below:
>
> Weaknesses:
> 1. We agree that we are not the first to explore using LLMs to augment CLIP training. However, our novel approach shows significant improvement over previous methods, including three appearing in CVPR 2023. The image-text matching aspect of our method, leveraging CLIP encodings, is novel compared to previous approaches. We agree that [1] and [2] are relevant, and cite them as concurrent work. In our revision, we will include additional discussions on how these two works relate to GIST and make sure that the differences are made clear. Briefly, GIST, unlike [1], does not require text captions in the dataset; rather we only require an image dataset with class names, and we generate our own text for fine-grained image datasets. It  would be interesting in future work to finetune our models using the pre-trained CLIP produced by [1]. [2] aims to improve accuracy for zero-shot CLIP. In contrast, our generated image-text pairs allow us to fine tune CLIP to achieve higher overall classification accuracy.
> 2. We include an ablation on the number of matched captions in Table 3 of the main paper. It suggests that matching more captions per image does not necessarily improve performance. The optimal number of captions per image varies per dataset.
> 3. We compare against CLIP LP and CLIP zero-shot and show that fine-tuning CLIP performs significantly better. We also show ablations for fine tuning CLIP on GIST captions versus fine tuning CLIP on captions from GIT and BLIP-2 or on standard text prompts (e.g.,the  FLYP method). We show that fine tuning on our GIST captions outperforms fine tuning CLIP on all of these alternative text approaches. We demonstrate that fine-tuning with the full dataset of images in comparison to few-shot numbers of images (Table 1,6) improves classification performance.
>
> Questions:
> 1. The text inputs to GIST are text prompts using class names. We show all of our text prompts in Appendix section A.3. We add material to the main text about prompting to clarify this.
> 2. In Section 4.5, we provide experimental results on fine tuning CLIP on short text descriptions versus fine tuning CLIP on the original descriptions. Both methods outperform the baselines, but fine tuning CLIP on short text descriptions does best. The text descriptions are used to finetune CLIP to improve the embedding space. No text descriptions are used to finetune the classifier.

---

### Meta-Review · Area_Chair_oKGq · 2023-12-07

**Metareview:**

The paper presents a method for generating fine-grained descriptions for images with a class label using an LLM and CLIP. Then the generated texts are used to finetune CLIP to increase its capability of understanding domain-specific attributes. The experiments in both full-shot and few-shot settings show improvements over previous approaches and the baseline CLIP.

Despite the performance improvements on different fine-grained classification benchmarks, all three reviewers raise concerns about the novelty of the method as there are many previous approaches for generating domain-specific data using LLMs and CLIP. Although the authors contrasted their method against the existing works, the distinctions do not appear to be notably significant.

Overall, the AC believes that the drawbacks of the paper outweigh its advantages, leading to the recommendation for rejection.

**Justification For Why Not Higher Score:**

All the reviewers share similar novelty concerns and the provided author rebuttal seems not very strong on this point.

**Justification For Why Not Lower Score:**

N/A

---

### Decision · Program_Chairs · 2024-01-16

Reject